# Unraveling Transcriptome Profile, Epigenetic Dynamics, and Morphological Changes in Psoriasis-like Keratinocytes: “Insights into Similarity with Psoriatic Lesional Epidermis”

**DOI:** 10.3390/cells12242825

**Published:** 2023-12-12

**Authors:** Ameneh Ghaffarinia, Szilárd Póliska, Ferhan Ayaydin, Aniko Goblos, Shahram Parvaneh, Máté Manczinger, Fanni Balogh, Lilla Erdei, Zoltán Veréb, Kornélia Szabó, Zsuzsanna Bata-Csörgő, Lajos Kemény

**Affiliations:** 1HCEMM-USZ Skin Research Group, H-6720 Szeged, Hungary; ameneh.ghaffarinia@med.lu.se (A.G.); shparvaneh79@gmail.com (S.P.); balogh.fanni@med.u-szeged.hu (F.B.); erdei.lilla@med.u-szeged.hu (L.E.); szabo.kornelia@med.u-szeged.hu (K.S.); bata.zsuzsa@med.u-szeged.hu (Z.B.-C.); 2Department of Dermatology and Allergology, Albert Szent-Györgyi Medical School, University of Szeged, H-6720 Szeged, Hungary; manczinger.mate@med.u-szeged.hu; 3Doctoral School of Clinical Medicine, University of Szeged, H-6720 Szeged, Hungary; 4Genomic Medicine and Bioinformatics Core Facility, Department of Biochemistry and Molecular Biology, Faculty of Medicine, University of Debrecen, H-4032 Debrecen, Hungary; poliska@med.unideb.hu; 5HCEMM-USZ Functional Cell Biology and Immunology, Advanced Core Facility, H-6728 Szeged, Hungary; ferhan.ayaydin@hcemm.eu; 6Institute of Plant Biology, Biological Research Centre, H-6726 Szeged, Hungary; 7Centre of Excellence for Interdisciplinary Research, Development and Innovation, University of Szeged, H-6720 Szeged, Hungary; goblos.aniko@szte.hu (A.G.); vereb.zoltan@med.u-szeged.hu (Z.V.); 8Regenerative Medicine and Cellular Pharmacology Laboratory (HECRIN), Department of Dermatology and Allergology, University of Szeged, H-6720 Szeged, Hungary; 9Systems Immunology Research Group, Institute of Biochemistry, HUN-REN Biological Research Centre, H-6726 Szeged, Hungary; 10HCEMM-Systems Immunology Research Group, H-6726 Szeged, Hungary; 11HUN-REN-SZTE Dermatological Research Group, H-6720 Szeged, Hungary

**Keywords:** psoriasis, keratinocyte, cytokine, inflammation, transcriptome, 5-methylcytosine (5-mC), 5-hydroxymethylcytosine (5-hmC), DNA methylation, DNA hydroxymethylation, epigenetic regulators

## Abstract

Keratinocytes are one of the primary cells affected by psoriasis inflammation. Our study aimed to delve deeper into their morphology, transcriptome, and epigenome changes in response to psoriasis-like inflammation. We created a novel cytokine mixture to mimic mild and severe psoriasis-like inflammatory conditions in cultured keratinocytes. Upon induction of inflammation, we observed that the keratinocytes exhibited a mesenchymal-like phenotype, further confirmed by increased *VIM* mRNA expression and results obtained from confocal microscopy. We performed RNA sequencing to achieve a more global view, revealing 858 and 6987 DEGs in mildly and severely inflamed keratinocytes, respectively. Surprisingly, we found that the transcriptome of mildly inflamed keratinocytes more closely mimicked that of the psoriatic epidermis transcriptome than the severely inflamed keratinocytes. Genes involved in the IL-17 pathway were a major contributor to the similarities of the transcriptomes between mildly inflamed KCs and psoriatic epidermis. Mild and severe inflammation led to the gene regulation of epigenetic modifiers such as HATs, HDACs, DNMTs, and TETs. Immunofluorescence staining revealed distinct 5-hmC patterns in inflamed versus control keratinocytes, and consistently low 5-mC intensity in both groups. However, the global DNA methylation assay detected a tendency of decreased 5-mC levels in inflamed keratinocytes versus controls. This study emphasizes how inflammation severity affects the transcriptomic similarity of keratinocytes to psoriatic epidermis and proves dynamic epigenetic regulation and adaptive morphological changes in inflamed keratinocytes.

## 1. Introduction

Psoriasis is a common chronic autoimmune skin disease affecting approximately 125 million people worldwide [1]. This condition arises from a complex interplay between immune system dysregulation and skin cell responses. Specifically, pro-inflammatory cytokines such as IL-17A, IL-22, TNF-α, and IFN-γ trigger an inflammatory cascade within the affected skin regions. In particular, keratinocyte cells (KCs), the primary cell type in the epidermal layer, respond robustly to these cytokines by releasing further inflammatory mediators, thereby perpetuating local skin inflammation. Consequently, this process leads to hyperproliferation and premature differentiation of KCs, resulting in psoriatic lesions. These lesions, characterized by well-demarcated, thickened, itchy, scaly, and inflamed patches of skin, commonly manifest on areas such as the elbows, knees, scalp, and lower back [2,3,4]. 

While several studies have extensively explored the effects of psoriasis-like inflammatory conditions on human epidermal KCs [5,6,7,8], most of these investigations utilized relatively high concentrations of cytokines to induce the inflammatory environment, potentially deviating from the in vivo conditions in the psoriatic skin [9,10,11,12]. Furthermore, comparative microarray analyses between the transcriptomes of psoriatic lesional skin and cytokine-stimulated cultured human epidermal KCs which were exposed to a range of cytokines with high concentrations of between 20 and 200 ng/mL [13], including, IL-17, IL-22 [9], IL-1α, IFN-γ [10], TNF-α [11], and oncostatin M (OSM) [12] revealed little overlap between the transcriptomes [13].

In parallel to transcriptome studies, epigenetic regulatory mechanisms, including DNA methylation and DNA hydroxymethylation, have been recognized for their pivotal role in modulating gene expression without altering the genomic sequence. These mechanisms can be significantly influenced by environmental factors, particularly inflammation [14]. Notably, alterations in the DNA methylation and hydroxymethylation epigenetic marks, 5-methylcytosine (5-mC) and 5-hydroxymethylcytosine (5-hmC) have been observed in psoriatic lesional skin, indicating their potential involvement in the disease pathogenesis [15,16,17,18]. However, there has been no investigation into the alterations of 5-mC and 5-hmC, as well as possible modifications in epigenetic regulators such as histone acetyltransferases (HATs), histone deacetylases (HDACs), histone methyltransferases (HMTs), histone demethylases (HDMs), DNA methyltransferases (DNMTs), and Ten methylcytosine dioxygenases (TETs), in human epidermal KCs in response to psoriasis-like inflammatory conditions. 

Therefore, this study aimed to address the effect of a psoriasis-like inflammatory milieu induced by a novel cytokine mixture (referred to as ‘CytoMix’) including IL-17A, IL-22, TNF-α, IFN-γ, and keratinocyte growth factor/fibroblast growth factor 7 (KGF/FGF7), on healthy human epidermal KCs at a lower, more in vivo-relevant cytokine concentrations (referred to as ‘mild inflammation’). By including KGF/FGF7 in addition to the aforementioned critical pro-inflammatory cytokines, our goal was to mimic the fibroblast-mediated keratinocyte stimulation seen in psoriatic lesions [19]. 

Our research group has already demonstrated the utility of xCELLigence Real-time Cell Analysis (RTCA) in studying how human keratinocytes respond to the inflammation triggered by bacterial stimuli and modifications in the extracellular matrix [20,21]. The RTCA is a live cell analysis system that provides a label-free environment for cells and enables tracking of cellular properties like proliferation, migration, cytotoxicity, adherence, and remodeling throughout the experiment [22]. This study represents our pioneering use of RTCA as a valuable technique for monitoring the influence of psoriasis-like inflammation induced by the CytoMix on the cellular fate of healthy human epidermal KCs. Another objective of applying RTCA was to determine the time point at which inflammation induced by the CytoMix had the most significant effects on the cellular properties of KCs. We selected samples from the time points identified by RTCA for further analysis. Alterations in tight junction (TJ) proteins, including claudins (CLDNs), have been associated with plaque-type psoriasis [23]. Notably, there is a confirmed correlation between RTCA results and the regulation of TJ proteins, especially Claudin-1 (CLDN1), observed in cultured keratinocytes [22]. CLDN1 is one of the major determinants of epidermal barrier functions in the human epidermis [24]. Considering these insights from prior research, our next aim was to investigate the relationship between RTCA results and the *CLDN1* gene expression regulation within psoriasis-like inflamed keratinocytes, comparing them to their respective controls.

Using confocal microscopy, we explored morphological changes in KCs following the induction of psoriasis-like inflammation and confirmed our observations using immunofluorescence staining and gene expression analysis. Furthermore, we have provided the transcriptome profile of KCs cultured under mild and 10× higher concentrations of CytoMix (referred to as ‘severe inflammation’). Our goal was to assess how closely the transcriptome of KCs resembles that of psoriatic epidermis under mild and severe psoriasis-like inflammatory conditions.

Moreover, we examined our transcriptome data to investigate changes in gene expression of epigenetic modifiers within mild and severe psoriasis-like inflammation. We also performed immunofluorescence staining and a global methylation assay to identify 5-mC/5-hmC profiles of psoriasis-like inflamed keratinocytes compared to controls. 

In summary, our research offers a deeper understanding of the morphological, transcriptomic, and epigenetic changes in healthy human epidermal KCs in mild and severe psoriasis-like inflammation.

## 2. Materials and Methods

### 2.1. Isolation and Culture of Skin Epidermal Keratinocyte Cells (KCs)

KCs were isolated from healthy abdominal skin samples obtained from the Plastic Surgery—Inpatient Care Unit, Department of Dermatology and Allergology, University of Szeged. Briefly, the epidermis was separated from the dermis using Dispase II solution treatment (2 U/mL, Roche Diagnostics, Basel, Switzerland) overnight at 4 °C [25,26]. The epidermal pieces were placed in a culture dish and incubated with trypsin for 15 min at 37 °C. Isolated KCs were cultured in keratinocyte serum-free medium (KSFM, Life Technologies, Carlsbad, CA, USA) supplemented with 1% antibiotic/antimycotic solution (AB/AM, Sigma Aldrich, St. Louis, MO, USA), 50 mg/mL brain pituitary extract (BPE, Life Technologies), and 5 ng/mL human recombinant epidermal growth factor (rEGF, Life Technologies) in a humidified incubator with 5% CO_2_ at 37 °C until around 70% confluency, then sub-cultured to passage number two or three. 

### 2.2. Cytokine Mixture (CytoMix) and Reagents

The CytoMix used in this study was a combination of animal-free human recombinant IL-17A (AF-200-17, PeproTech, Cranbury, NJ, USA), animal-free human recombinant IL-22 (AF-200-22, PeproTech, NJ, USA), animal-free human recombinant TNF-α (AF-300-01A, PeproTech, NJ, USA), animal-free human recombinant IFN-γ (AF-300-02, PeproTech, NJ, USA), and recombinant KGF/FGF7 (Cyt-219, Prospec Bio, Rehovot, Israel). To prepare the CytoMix, the cytokines were first dissolved in injection-grade water and then diluted in 0.1% bovine serum albumin (BSA, PM-T1725/500, Biosera, Cholet, France) in sterile 1× PBS. The resulting mixture was aliquoted and stored at −80 °C until use. The cytokines were mixed in basic KSFM with L-glutamine, excluding AB/AM and supplements (BPE and EGF). The specific combinations and concentrations of pro-inflammatory cytokines and the growth factor were carefully selected based on pilot experiments. We used an identical mixture with the cytokines and the growth factor at 10× higher concentrations to induce severe inflammation.

### 2.3. xCELLigence Real-Time Cell Analysis (RTCA)

The RTCA system (xCELLigence RTCA SP, ACEA Biosciences, San Diego, CA, USA) utilizes an E-plate 96 well, featuring sensor electrodes for signal induction, to primarily detect changes in impedance. Cell attachment or detachment from the surface electrodes of the E-plate 96 well induces electronic impedance, with variations in cell shape, proliferation, and adhesion rate generating distinct impedances over time. These impedance signals are transformed into a cell index (CI), represented as colorful curves, offering a visual diagram of the biological status of cells. A CI of zero indicates the absence of cells, while higher values reflect increased cell adhesion to the plate [27]. 

We plated 3.0 × 10^4^ cells/well (passage number 3 or 4) in 150 µL supplemented (EGF and BPE) KSFM with L-glutamine but without AB/AM [20] in fibronectin-coated (35 µL of 4 µg/mL, F1141-1MG, Sigma) E-plates 96 well. After 24 h, the cells were quickly washed twice with 1× PBS, and EGF and BPE starved for an additional 24 h before CytoMix stimulation. We configured the RTCA software (version 2.0) to record the CI values every 15 min. However, upon exporting these values, we employed CI and normalized cell index (nCI) data at 4 h intervals to recreate the exact plots automatically generated by the RTCA software, this time using Prism software 8.0.2. The nCI values were determined by selecting a specific time (e.g., when cells were stimulated with CytoMix) and setting it as the reference point (1.0 or 100%). All other values are expressed as proportions relative to this reference point. Data were collected from three distinct biological replicates (*n* = 3), and each condition was assessed with three technical replicates.

### 2.4. Total RNA Isolation and Real-Time RT-PCR

For real-time RT-PCR analysis, total RNA was extracted from cultured KCs using TRI-Reagent (Molecular Research Center; Cincinnati, OH, USA). The purity and concentration of the RNA samples were assessed using a Nanodrop spectrometer (Colibri Mikrovolüm Spektrometre, Berthold, Germany). Qualified RNA samples were then utilized for cDNA synthesis with the UltraScript cDNA Synthesis Kit (Thermofisher, Waltham, MA, USA). Subsequently, real-time RT-PCR was performed using TaqMan assays (Thermofisher, Waltham, MA, USA) and the qPCRBIO Probe Mix Lo-ROX kit (PCR Biosystem Ltd., London, UK) on a C1000 Touch Thermal Cycler (Bio-Rad Laboratories, Hercules, CA, USA). Assay numbers are listed in a Appendix A. All genes were normalized to the *18S* rRNA gene, and the relative mRNA expression levels were calculated using the ΔΔCt method, compared with the time-matched unstimulated control samples as the reference. Each real-time RT-PCR reaction was performed in duplicates.

### 2.5. IL-8 ELISA Assay

We evaluated the level of cell activation triggered by mild (1×) and severe (10×) CytoMix by measuring the secretion of IL-8 (CXCL8) protein in the culture supernatants obtained from KCs exposed to these treatments for 48 h. Quantification of IL-8 production was carried out using a Human IL-8 (CXCL8) Mini TMB ELISA Development Kit (900-TM18, PEPROTECH) following the provided manufacturer’s instructions.

### 2.6. High-Throughput RNA Sequencing

A high-throughput RNA sequencing analysis was performed on the Illumina sequencing platform to obtain global transcriptome data. Total RNA sample quality was checked on the Agilent BioAnalyzer using the Eukaryotic Total RNA Nano Kit according to the manufacturer’s protocol. RNA samples, comprising twelve in total (three controls and three CytoMix-treated for mild inflammation, as well as three controls and three CytoMix-treated for severe inflammation), were selected based on RNA integrity number (RIN value) ranging from 8.9 to 9.8. These samples were chosen to undergo library preparation for RNA sequencing. RNA sequencing libraries were prepared from total RNA using an Ultra II RNA Sample Prep kit (New England BioLabs, Ipswich, MA, USA) according to the manufacturer’s protocol. Oligo-dT conjugated magnetic beads captured poly-A RNAs, and the mRNAs were then eluted and fragmented at 94 °C. First-strand cDNA was generated through random priming reverse transcription and double-stranded cDNA was developed after the second-strand synthesis step. After repairing ends, A-tailing, and adapter ligation steps, adapter-ligated fragments were amplified in enrichment PCR; finally, sequencing libraries were generated. Sequencing runs were executed on the Illumina NextSeq 500 instrument using single-end 75-cycle sequencing.

Raw sequencing data (fastq) were aligned to human reference genome version GRCh38 using the HISAT2 algorithm, and BAM files were generated. Downstream analysis was performed using Strand NGS software version 4.0—Build 242089 (https://www.strand-ngs.com/ (accessed on 1 June 2023)). BAM files were imported into the software, and the DESeq algorithm was used for normalization. A moderated *t*-test was used to determine differentially expressed genes (DEGs) in mildly and severely inflamed KCs versus control groups. Genes with *p* < 0.05 and fold change (FC) ≥ 2 were considered DEGs. Raw sequencing data is available in the NCBI under the BioProject ID: PRJNA1047331.

Cytoscape v3.4 software with ClueGo v2.3.5. application was used for identifying over-represented gene ontology (GO) terms. A two-sided hypergeometric test with Benjamini–Hochberg FDR correction was performed using the list of DEGs and the GO biological process database. Protein-protein interaction (PPI) networks were generated with STRING v12.0 using default parameters and at least medium confidence interactions.

### 2.7. Examining the Transcriptome Similarity between Mild and Severely Inflamed KCs and Psoriatic-Lesional Epidermis

Gene expression data of psoriatic and healthy epidermis samples were downloaded from the Gene Expression Omnibus (series matrix files of GSE68937 and GSE68923 datasets). GSE68937 and GSE68923 datasets contain the microarray results from two models, including three lesional and two healthy epidermis samples in set one and three lesional and three healthy epidermis samples in set two [28]. Wilcoxon’s rank-sum test was used to identify the DEGs between lesional and healthy epidermal samples. The *p* values were adjusted according to Benjamini and Hochberg [29], and genes with a false discovery rate lower than 0.1 (FDR < 0.1) were classified as DEGs. IDs were collated using Human Gene Nomenclature Organization (HUGO) website data. Fold change values were calculated by dividing the median gene expression values in psoriatic samples by the ones in healthy samples. The overlapping gene function was identified with a GO enrichment analysis using the GOrilla software [30,31]. Kyoto Encyclopedia of Genes and Genomes (KEGG) pathway enrichment analysis was conducted using the WebGestalt online tool. The classification of genes was the same as for the study with GOrilla. Pathways with an FDR < 0.2 were considered significantly enriched. R program (version R 3.6.3) was used for statistical analyses.

### 2.8. Immunofluorescence Staining

KCs were seeded at a density of 1.0–1.2 × 10^5^ cells per well in 8-well slide chambers. To perform 5-mC/5-hmC immunofluorescence staining, after 48 h of mild inflammation induction, the cells were fixed with 4% paraformaldehyde (PFA) for 10 min at room temperature (RT). Then, the cells were washed for 5 min with 1× PBS and permeabilized with 0.1% Triton X-100 in 1× PBS for 10 min at RT. For 5-mC and 5-hmC co-localization, antigen retrieval was carried out by incubating the cells for 1 h with freshly made 2 N hydrochloric acid (HCl) in 1× PBS at RT. After DNA denaturation with HCl, the cells were neutralized using 0.1 M Tris-HCl (pH 8.3) for 10 min. Unspecific binding sites were blocked by incubating the cells in a blocking solution (1% normal goat serum (NGS)) and 1% BSA in 1× PBS for 30 min at RT. The KCs were incubated overnight at 4 °C with the following primary antibodies: rabbit-polyclonal anti-human 5-hmC (1:1000, Active motif, Cat. No. 39769, Carlsbad, CA, USA) and mouse-monoclonal anti-human 5-mC (1:500, Epigentek, Cat. No. A-1014, East Farmingdale, NY, USA). For vimentin (VIM) immunofluorescence staining, the cells were fixed and permeabilized with ice-cold methanol-acetone (1:1) for 5 min [32]. Then, the cells were air-dried for 1 min and incubated for 30 min with the same blocking solution we used for 5-mC/hmC staining. Subsequently, the cells were incubated overnight at 4 °C with rabbit-polyclonal anti-human VIM antibody (1:200, Abcam, Cat. No. ab137321, Cambridge, UK). AlexaFluor 546-conjugated anti-rabbit IgG (1:500, Life Technologies, Carlsbad, CA, USA) and AlexaFluor 488-conjugated anti-mouse IgG (1:500, Life Technologies, Carlsbad, CA, USA) were used as secondary antibodies. The nuclei were visualized using DAPI staining (Sigma-Aldrich, St. Louis, MO, USA).

### 2.9. Confocal Microscopy 

Images were acquired using a confocal laser scanning microscope Leica STELLARIS 5 (Leica Microsystems CMS GmbH, Germany) by using 405 nm and 568 nm lasers for excitation of DAPI and RFP with emission filters of 410–512 nm and 573–750 nm, respectively. Images were taken with an HC PL APO CS2 20× (N.A. 0.75) DRY objective, unidirectional scanning mode, and were processed by Leica Application Suite version X (LAS-X, 4.4.0.24861) software. An identical microscope configuration was used for all intensity comparison images.

### 2.10. DNA Isolation and Quantification of Global DNA Methylation Levels

The cells were cultured at a cell density of 2.5–3.0 × 10^5^ cells/well (passage number 3 or 4) in 12-well culture plates using supplemented KSFM containing L-glutamine, without AB/AM. After 48 h of mild and severe inflammation induction, the cells were washed with 1× PBS-containing EDTA (PBS-EDTA) and detached using trypsin. Then, cells were collected in PBS-EDTA and centrifuged twice at 500× *g* for 5 min at 4 °C. The resulting cell pellet was resuspended in 50 µL of DNA/RNA shield (ZYMO RESEARCH, R1100-50). Genomic DNA was extracted with Quick-DNA^TM^ Miniprep Plus Kit based on the manufacturer’s instruction (ZYMO RESEARCH, D4068). The purity and concentration of the DNA samples were determined using a Nanodrop (Colibri Mikrovolüm Spektrometre, Berthold, Germany). We used the Global DNA Methylation Assay Kit (5-Methyl Cytosine, Colorimetric) (Abcam, ab233486, Cambridge, UK) according to the manufacturer’s instructions to assess 5-mC% in total DNA. In each reaction, 100 ng of DNA with a 260/280 ratio > 1.6 was used.

### 2.11. Statistical Analysis and Graphical Abstract

GraphPad Prism 8.0.2 (GraphPad Software, San Diego, CA, USA) was used both for conducting statistical analyses, by applying Student’s unpaired *t*-test, and for generating graphical representations of the xCELLigence RTCA, real-time RT-PCR, and ELISA results. The graphical abstract was created using Biorender.com (accessed on 8 November 2023).

## 3. Results

### 3.1. The Real-Time Cell Analysis (RTCA) Confirms the Significant Impact of Mild CytoMix Treatment on the Cellular Fate of KCs

Within the first 24 h post-CytoMix treatment, no differences in the nCI values were observed when comparing the CytoMix-treated KCs and controls. However, from 32 h post-inflammation induction, the CytoMix-treated cells consistently showed significantly lower nCI values than their respective controls (Figure 1A). At the end of the RTCA experiment, we observed the KCs cultures under a transmitted light microscopy to characterize possible morphological changes in response to the CytoMix treatment. Microscopic analysis revealed distinct cellular patterns in control and CytoMix-treated KCs. Confluent cells in both control and CytoMix-treated groups formed a stratified arrangement, with at least one layer of KCs positioned above the other. However, the CytoMix-treated KCs exhibited EMT-like morphology [33], characterized by a spindle shape with elongated intercellular connections. In contrast, cells in the control group showed a more regular, polygonal shape (Figure 1B,C). Few regions in both control and CytoMix-treated KCs only had one layer of KCs, suggesting a more regular, monolayer arrangement. 

We subsequently investigated the relationship between *CLDN1* gene expression regulation and the lower CI/nCI values we observed in CytoMix-treated KCs compared to their respective controls. Utilizing real-time RT-PCR, we assessed the mRNA expression level of the *CLDN1* gene. Our analysis revealed a significant decrease in *CLDN1* mRNA expression in CytoMix-treated KCs compared to controls at 48 and 72 h post-inflammation induction, as illustrated in Figure 1D. This finding suggests that the treatment with CytoMix induces changes in *CLDN1* gene expression that can potentially modify the keratinocyte barrier status, resulting in lower CI/nCI values in psoriasis-like KCs compared to their controls. 

### 3.2. Morphological Changes and Increased Vimentin (VIM) Expression Suggest an Epithelial-Mesenchymal Transition (EMT)-like Phenomenon in Mild CytoMix-Treated KCs

In light of the observed morphological changes resembling EMT in our transmitted light microscopy images of E-plates 96 well, we decided to perform immunostaining for the VIM protein and assess the level of *VIM* gene expression as a marker of the EMT process. 

The confocal microscopy analysis confirmed the presence of elevated VIM protein levels in CytoMix-treated KCs compared to controls in the 48 h samples (Figure 2A,B). This difference is evident in the top layer panels, while there is no clear distinction in VIM expression between the bottom layers of both groups. The morphology of the spindle shape with long-armed intercellular connections in CytoMix-treated KCs and the polygonal shape in control KCs is shown in Figure 2C. Using real-time RT-PCR, we detected increased *VIM* mRNA expression levels in CytoMix-treated KCs compared to the controls at 48 and 72 h post-treatment (Figure 2D).

These results suggest that CytoMix-treated KCs partially transitioned from an epithelial to a more mesenchymal state, resembling the characteristics of psoriatic epidermal cells [34].

### 3.3. Mild CytoMix Treatment Induces Cytokeratin Gene Expression Changes

Given these morphological changes, we set out to determine if gene expression of cytokeratin genes known to be dysregulated in psoriatic lesional epidermis was also affected. These genes included keratin *(KRT)1* and *KRT10,* which play a role in differentiation [35]; *KRT15,* representing cell activation [36]; and *KRT17,* which is important during the proliferation processes of keratinocytes [37] (Figure 3). 

Our investigation revealed a significant down-regulation of *KRT1* and *KRT10* in CytoMix-treated KCs compared to their controls at 24, 48, and 72 h time points. Exploring KRT15 mRNA expression, we noted transient changes: a significant down-regulation in CytoMix-treated KCs at 24 and 48 h and increased expression at 72 h compared to untreated control cells. 

For KRT17 mRNA expression, an up-regulation was observed in CytoMix-treated KCs compared to controls, although without statistical significance.

### 3.4. Severely Inflamed KCs Produce Significantly Higher Levels of Pro-Inflammatory Cytokines Than Mildly Inflamed Ones

To test our hypothesis that treating KCs with 10× higher levels of the CytoMix results in a more potent or severe inflammatory response, we assessed secreted IL-8 (CXCL8) protein and *IL-23A* mRNA levels known to exhibit elevated expression in KCs within psoriatic lesions [38,39]. We observed notably elevated IL-8 levels in both mildly and severely inflamed KCs compared to their controls, with more pronounced changes in the severely inflamed group. Simultaneously, we analyzed the mRNA expression level of *IL-23A* and found that its expression markedly increased in severely inflamed KCs compared to their controls. In contrast, mildly inflamed KCs exhibited modest changes (Figure 4). Interestingly, the morphological appearance of both mildly and severely inflamed KCs was similar despite the observed differences in their cytokine production (Appendix A). Our findings clearly show increased inflammatory responses in KCs subjected to 10× CytoMix treatment.

### 3.5. 10× CytoMix-Treatment Results in a Higher Number of Differentially Expressed Genes (DEGs)

Based on our previous results, we performed RNA sequencing using the 48 h CytoMix-treated samples. This analysis revealed more DEGs for severely than mildly inflamed KCs compared to their controls (Table 1). Among the 858 identified DEGs, 285 (33.21%) were down-regulated, and 573 (66.79%) were up-regulated in mildly inflamed KCs compared to the control samples. In contrast, of the 6987 DEGs in severely inflamed KCs, 3655 (52.31%) were identified as down-regulated, and 3332 (47.69%) as up-regulated genes. Based on these results, the number of up-regulated DEGs in mildly inflamed KCs was almost twice that of the down-regulated DEGs in the same group when compared to their controls. Heatmap analysis shows the gene expression pattern of mildly and severely inflamed KCs compared to their controls (Figure 5). 

### 3.6. Gene Ontology (GO) Enrichment Analysis of the DEGs of Mildly and Severely Inflamed KCs

We performed a comprehensive Gene Ontology (GO) enrichment analysis on the DEGs obtained from mildly and severely inflamed KCs to uncover the most significant and highly enriched biological processes (BPs), cellular components (CCs), and molecular functions (MFs) that underwent differential regulation in inflamed KCs vs. controls. In comparing mildly inflamed KCs and their controls (Figure 6), the analysis revealed significant differences in 97 BPs, 32 CCs, and 9 MFs (*p* < 0.05). On the other hand, in the comparison between severely inflamed KCs and their control group (Figure 7), there were notable differences in 107 BPs, 53 CCs, and 57 MFs (*p* < 0.05). Furthermore, based on the enrichment in terms, we identified the top five pathways in each of the three BP, CC, and MF categories. In mildly inflamed KCs, within the top pathway in the BPs (tissue development), the CCs (cytoplasmic vesicle part), and the MFs (interleukin-1 receptor binding), there were 167, 119, and 7 DEGs, respectively. This suggests that in mild inflammation, the primary impact is on genes associated with tissue development and immune responses. For severely inflamed KCs vs. their controls, there were 2536, 5190, and 42 DEGs, respectively, in the top BP (cellular nitrogen compound metabolic process), CC (intracellular part), and MF (cation channel activity) categories (Appendix A). These propose that in severe inflammation, there is a complex response affecting metabolism, cellular organization, and ion channels. 

Table 2 presents the top 20 genes displaying the greatest expression changes in our dataset. In mildly inflamed KCs, the most up-regulated gene was S100 Calcium Binding Protein A7A (*S100A7A*), which is also known as koebnerisin (FC: +4453.61), while keratin 1 (*KRT1*) showed the highest down-regulation (FC: −151.88). In severely inflamed KCs, defensin beta 103 (*DEFB103*) exhibited the highest up-regulation (FC: +8680.30), and inhibitor of DNA binding 3 (*ID3*) showed the most notable down-regulation (FC: −571.72). Among the 20 most down-regulated DEGs of mildly and severely inflamed KCs, 4 were common, including *KRT1*, *EPHB6*, *CCDC3*, and *PALMD*. Decreased levels of the *KRT1* [40] and EPH receptor B6 *(EPHB6)* [41] have been seen in psoriatic lesions. However, Coiled-coil domain containing 3 (*CCDC3*) and palmdelphin (*PALMD*) have not been reported as known factors contributing to the development or progression of psoriasis. Between the 20 most up-regulated DEGs in mildly and severely inflamed KCs, 12 genes were found to be shared. These genes included S100 calcium-binding protein A7 (*S100A7*), also known as (psoriasin), defensin beta (*DEFB)4A* and *DEFB4B*, late cornified envelope *(LCE)3A*, *LCE3E*, *LCE3D*, small proline-rich proteins *(SPRR)2C*, *SPRR2A*, *SPRR2F*, *SPRR2G*, Carcinoembryonic antigen-related cell adhesion molecule 6 (*CEACAM6*), and Chromosome 15 Open Reading Frame 48 (*C15orf48*), all of which exhibit increased expression in psoriatic lesional epidermis [42,43,44,45,46,47,48,49]. The up-regulated shared genes mainly play roles in the terminal differentiation and cornification processes (*S100A7*, *LCE3A*, *LCE3E*, *LCE3D*, *SPRR2C, SPRR2A*, *SPRR2F*, *SPRR2G*) and antimicrobial defense (*DEFB4A*, *DEFB4B*) of KCs. 

Furthermore, we explored the possible protein–protein interactions (PPI) among the 12 shared genes using the STRING database and showed that they are involved in late epidermal differentiation and cornified envelope formation (Figure 8). 

### 3.7. Dynamic Epigenetic Alterations in Psoriasis-like Mildly and Severely Inflamed Keratinocytes (KCs)

In the transcriptome analysis of mildly and severely inflamed KCs, we detected significant alterations in the gene expression of critical epigenetic modifiers (Table 3). Among HATs, *HAT1* transcript, a cytoplasmic HAT (type B) [50], was significantly down-regulated in severely inflamed KCs. Expression of *HDAC9* transcript, a member of the histone deacetylase family [51], was significantly up-regulated in mild and severe inflammatory conditions. In contrast, no significant differences were observed within the HMTs and HDMs families. 

*DNMT1*, responsible for maintaining DNA methylation patterns, was significantly down-regulated in mildly and severely inflamed KCs. Within the TETs family, only *TET3* displayed significant up-regulation in severely inflamed KCs compared to controls. Notably, *TET3* and *DNMT3A* exhibited distinctive fold change directions in severely inflamed KCs compared to mildly inflamed ones. Specifically, these two genes were down-regulated in mildly inflamed KCs but showed significant up-regulation in the severely inflamed KCs. This contrasting regulation may suggest the dynamic and complex nature of epigenetic changes during different stages of inflammation in psoriasis.

We also performed real-time RT-PCR experiments to validate the NGS data for selected candidate transcripts, including *HDAC9*, *DNMTs*, and *TET3*. While the fold change directions were aligned with the transcriptomic data for most of the chosen transcripts, the significant differences were not consistently confirmed through real-time RT-PCR (Figure 9). Furthermore, through real-time RT-PCR, we demonstrated that *DNMT3B* was also significantly down-regulated in mildly and severely inflamed KCs, with a more pronounced down-regulation observed in severely inflamed ones.

### 3.8. The Transcriptome of Mildly Inflamed Keratinocytes More Closely Mimicked That of the Psoriatic Epidermis Transcriptome Than the Severely Inflamed Keratinocytes

When comparing the psoriatic lesional and the healthy epidermis, 4104 DEGs were detected. Subsequently, the DEGs from the psoriatic lesional epidermis were compared with 858 DEGs from mildly inflamed KCs and 6987 DEGs from severely inflamed KCs. Among the 858 DEGs in mildly inflamed KCs, 157 genes overlapped. Moreover, 140 (89%) of the 157 overlapped genes showed the same fold change direction as the psoriatic lesional epidermis. Notably, the degree of change strongly correlated between the two comparisons (Spearman’s rho: 0.74, *p* < 2.2 × 10^−16^); the data are shown in Figure 10A. In the case of the 6987 DEGs in severely inflamed KCs, we found a higher number of overlapping genes (851), and 370 genes (43%) of the 851 overlapped genes showed the same fold change direction as the psoriatic lesional epidermis. However, the degree of change only weakly correlated (Spearman’s rho: 0.24, *p* = 7 × 10^−13^); the data are shown in Figure 10B.

### 3.9. The IL-17 Signaling Pathway Is Enriched in the Overlapping Gene Set of the Mildly Inflamed Keratinocytes and Psoriatic Lesional Epidermis

To identify pathways induced by our mild CytoMix and shared with those in psoriasis, we conducted a KEGG pathway analysis. This involved analyzing the 157 genes that exhibited overlap between the DEGs of mildly inflamed KCs and those in psoriatic lesional epidermis. We found that several of these transcripts (e.g., *DEFB4A*, *IL1B*, *LCN2*, *PTGS2*, *S100A7*, *S100A8*, and *S100A9*) represented the IL-17 signaling pathway (FDR ≤ 0.05, *p*-value: 0.00001), as shown in Figure 11. 

### 3.10. Exploring the Changes of 5-methylcytosine (5-mC) and 5-hydroxymethylcytosine (5-hmC) Epigenetic Marks in the Psoriasis-like Mild Inflammation

Our next aim was to explore the epigenetic profile of mildly inflamed KCs vs. their controls. Using confocal microscopy, we observed a weak 5-mC signal in both control and CytoMix-treated KCs (Figure 12A,B). However, the global DNA methylation assay identified a tendency of reduced 5-mC levels in both mildly and severely inflamed KCs when compared to the control group (Figure 12D). In the CytoMix-treated group, the top layer KCs exhibited a significantly stronger 5-hmC signal than the top layer of the control group. Conversely, the control group showed a relatively heightened 5-hmC signal in the bottom layer compared to the same layer of the CytoMix-treated group (Figure 12A, B). In certain regions of the samples with a single-layer arrangement, we found that the levels of 5-mC and 5-hmC signals were overall lower in CytoMix-treated KCs compared to the control group (Appendix A). 

Next, we assessed the mRNA expression levels of *TET1*, *TET2*, and *TET3* enzymes in mildly inflamed KCs versus their controls (Figure 12C). With real-time RT-PCR, we found that the *TET1* and *TET2* mRNA expression was significantly lower in CytoMix-treated KCs versus their controls at 24, 48, and 72 h post-inflammation induction. However, the *TET3* mRNA expression was only significantly decreased at 24 h post inflammation induction. 

## 4. Discussion

In this study, we used in vitro cultures of healthy human epidermal KCs as a model to investigate KC morphological changes, their transcriptomic profile, and the epigenetic regulators/marks alterations in response to psoriasis-like inflammation induction. By treating epidermal KCs with the key pro-inflammatory cytokines associated with the disease, such as IL-17A, IL-22, IFN-γ, and TNF-α [3], we intended to mimic the effects induced by the corresponding immune cells, which are not present in the KCs in vitro models. Moreover, prior studies discovered that activated lymphocytes induce fibroblasts in psoriatic lesions to generate KGF, driving keratinocyte hyperproliferation [19]. By adding KGF to the above-mentioned key pro-inflammatory cytokines, we aimed to mimic fibroblast-driven keratinocyte stimulation seen during disease pathogenesis. Therefore, we introduced a novel combination of cytokines, including IL-17A, IL-22, IFN-γ, TNF-α, and KGF as a CytoMix, that had not been previously applied for inducing psoriasis-like inflammation. To the best of our knowledge, our study is the first to give essential resources on the effects of psoriasis-like inflammation induced by pro-inflammatory cytokines on the cellular fate of human epidermal KCs using the RTCA platform. This can serve as a basis for future comparative studies. Using the RTCA system, we observed a significant decrease in nCI values for psoriasis-like inflamed KCs at 48 h post-inflammation induction compared to the control group. This observation influenced our decision to select the 48 h samples for our further studies. Additionally, the reduction in nCI values in psoriasis-like KCs correlated with a significant down-regulation of *CLDN1* gene expression, suggesting an induced epidermal barrier dysfunction in KCs by psoriasis-like inflammation. Nevertheless, it is important to note that the downregulation of *CLDN1* gene expression is most likely not the only factor contributing to the lower CI/nCI values observed in psoriasis-like KCs. On the other hand, the RTCA technique alone cannot precisely identify which specific cellular properties correspond to alterations in CI/nCI values; rather, it provides a general perspective on cellular changes during the experiment. Therefore, additional analyses are essential to determine the primary cellular properties responsible for the observed CI/nCI value changes. This aspect goes beyond the main objective of the current study and remains open for future research. Furthermore, our study indicated that psoriasis-like KCs underwent an EMT process, resulting in the acquisition of mesenchymal characteristics compared to their control group. This was evidenced by their altered morphology, such as spindle-shaped appearances with elongated intercellular connections and increased VIM levels. Although VIM is typically associated with mesenchymal cell types, it is noteworthy that it can also be present in non-mesenchymal cells like cultured KCs [32]. However, the EMT-like phenotype observed in KCs treated with the CytoMix, coupled with the increased expression of VIM, which is also seen in psoriatic lesional epidermis [34], strengthens the hypothesis of an EMT-like process occurring in the CytoMix-treated KCs. Other studies have also shown that treating primary human skin KCs with TNF-α or TGF-β induces mesenchymal morphological characteristics, similar to those we observed in our CytoMix-treated KCs, and is associated with increased VIM expression at both the gene and protein levels [33]. While our findings provide novel insights into the potential involvement of EMT-related pathways in the pathogenesis of psoriasis, further investigations are needed to elucidate the specific molecular events and signaling pathways orchestrating the observed EMT-like changes in CytoMix-treated KCs. Additionally, exploring the functional implications of these alterations and their relevance to psoriatic pathophysiology will be essential for a comprehensive understanding of the role of EMT in psoriasis.

This study also demonstrated that the CytoMix treatment mimics psoriasis-like effects by inducing changes in disease-associated cytokeratin genes in CytoMix-treated KCs compared to their controls. Furthermore, it highlights that *KRT1* and *KRT10* are the most prominently down-regulated keratin genes in response to CytoMix treatment. This underscores an aberrant epidermal differentiation pattern resulting from psoriasis-like inflammation, a characteristic feature of psoriatic lesional epidermis [52]. The decrease in *KRT15* expression maintains KCs in an activated state [36], a phenomenon also observed in psoriatic lesional epidermis, where there is a reduction in *KRT15*-expressing KCs [18]. Furthermore, we noted a rapid rise in mRNA levels of *KRT15* in CytoMix-treated KCs 72 h after inducing inflammation. This observation could indicate either the deactivation of the inflamed KCs [36] or a shift towards the characteristic KRT15 expression pattern found in basal layer KCs [17]. In our experiments, KCs were confluent and contact-inhibited before the inflammation induction. This condition may have promoted KC differentiation and the formation of stratified cultures, prohibiting a proliferative phenotype. Conducting experiments with lower cell densities may have induced more proliferative states than the confluent cultures, enabling an assessment of KRT17, which promotes hyperproliferation of KCs under inflammatory conditions and is known to be highly expressed in psoriatic lesional epidermis [37]. 

Furthermore, we used both low and high concentrations of the cytokines in our mixture to study the transcriptome of mildly and severely inflamed KCs. To analyze the transcriptome of psoriasis-like inflamed KCs, we employed RNA sequencing, which can detect a higher percentage of differentially expressed genes, especially those with low expression, in comparison to the microarrays used in previous studies [13]. We found a higher number of DEGs in severely inflamed KCs than in mildly inflamed ones compared to their controls. This finding emphasizes the pivotal role of CytoMix dosage in driving the extent of inflammation and subsequent gene expression changes. 

We recognized that the 20 prominently dysregulated genes in both mildly and severely inflamed KCs mainly belonged to the epidermal differentiation complex (EDC) gene cluster, primarily associated with epidermal differentiation and the formation of the cornified envelope. These disruptions ultimately contribute to a compromised skin barrier, a characteristic feature of psoriatic lesional skin [52]. 

We also observed that severely inflamed KCs exhibited a greater number of overlapping genes with the psoriatic lesional epidermis in comparison to mildly inflamed keratinocytes. However, the extent of change in gene expression showed a weak correlation in severely inflamed KCs, whereas a significant correlation was observed in mildly inflamed KCs. This highlights that subjecting the human epidermal KCs with 10× CytoMix or severe inflammation may create an environment less comparable with the inflammatory conditions found in psoriatic lesions. The overlapping genes of mildly inflamed KCs were mainly enriched in the IL-17 signaling pathway. This pathway plays a central role in the pathogenesis of psoriasis, contributing to the chronic inflammation and characteristic skin manifestations associated with the condition [53]. This insight highlights the potential significance of the IL-17 pathway as a therapeutic target in managing the disease’s complex pathogenic processes. Additionally, it demonstrates our model’s capability to accurately replicate and target this crucial pathway.

In addition, the transcriptomic profiles of mildly and severely inflamed KCs exhibited alterations in the mRNA expression levels of epigenetic modifiers, including *HAT1*, *HDAC9*, *DNMTs*, and *TETs*. The concurrent down-regulation of *HAT1* and up-regulation of *HDAC9* suggest a shift towards reduced histone acetylation, increased histone deacetylation, chromatin compaction, and potentially suppressed gene expression [54]. *DNMT1* acts as the core machinery for maintaining DNA methylation [55], and its decreased levels in CytoMix-treated KCs vs. their controls may be reflective of (passive DNA-demethylation) processes. On the other hand, decreased expression of *TET1* and *TET2* might have caused decreased 5-hmC levels, potentially leading to elevated 5-mC levels. TET enzymes play a crucial role in DNA demethylation by oxidizing 5-mC, leading to its conversion into 5-hmC, which is a step towards active DNA demethylation [56]. However, considering the reduced expression of *DNMT1*, it is plausible that both 5-mC and 5-hmC levels could be lower in CytoMix-treated KCs compared to their controls. *DNMT3B* expression is positively correlated with epidermal differentiation, as it is minimally expressed in undifferentiated basal layer KCs and actively being turned on during the differentiation process in the upper layer of a healthy epidermis [57]. Hence, the significant downregulation of *DNMT3B* in CytoMix-treated KCs implies that the mildly and severely inflamed cells might also be epigenetically associated with an undifferentiated state. *DNMT3A* and *TET3* also exhibited deregulation in mild and severe inflammatory conditions. However, their fold change directions were opposite in these two inflammatory conditions. This highlights the potential variations in the epigenetic regulation of certain modifiers between mild and severe inflammatory conditions. Collectively, our study confirms the profound impact of short-term inflammation, revealing its ability to trigger gene regulation in key epigenetic modifiers.

With immunofluorescence staining, we found distinct 5-hmC patterns in inflamed KCs versus controls, accompanied by consistently low 5-mC intensity in both groups. This observation suggests that certain common internal/external factors may have the potential to influence the 5-mC epigenetic mark, irrespective of the inflammatory milieu. Nevertheless, the Global DNA methylation assay further detected a tendency of decreased 5-mC levels in mildly and severely inflamed KCs compared to controls. The observed alterations in the 5-mC and 5-hmC profiles suggest a dynamic nature of epigenetic responses under psoriasis-like inflammatory conditions. 

In addition to these findings, it is essential to acknowledge the limitations of our study. In our experiments, the KCs were separated from the abdominal skin of healthy donors, and we did not examine whether the KCs from other body regions, such as the breast or thigh, would respond to the psoriasis-like inflammatory environment similarly. Our donor cohort consisted of five Hungarian female individuals, resulting in a homogeneous sample in terms of both ethnicity and sex. While this homogeneity provided a consistent baseline for our analysis, it is important to note that our study did not include KCs derived from male abdomen skin samples or individuals from other ethnicities. Therefore, we cannot determine how the response of male abdomen-derived KCs or other ethnicities compares to that of female abdomen-derived KCs. Regarding the importance of ethnicity, a study has highlighted its importance in skin responses to pro-inflammatory cytokines, as demonstrated by differential responses observed in 3D African American and White non-Hispanic skin organoids [58]. We also highlight that our transcriptomic data represents a particular time-point (48 h) after inflammation induction; thus, it cannot assay for dynamic changes and does not fully capture all aspects of the disease transcriptome. 

Our study’s strength lies in our analysis of transcriptomic profiles from mildly and severely inflamed KCs, originating from the same biological donors. This approach facilitated a direct comparison of gene expression signatures between mildly and severely inflamed keratinocytes and their alignment with the gene expression patterns found in psoriatic lesional epidermis. Importantly, this method allowed us to control for potential donor-related variations in both mild and severe inflammatory conditions. Furthermore, all our experiments were conducted in AB/AM-free culturing media. This aimed to minimize the influence of antibiotics and antimycotics on KCs responses to inflammation, thus better matching the natural conditions of the human body. 

All these emphasize the importance of considering such variables when studying the cellular fate of KCs in the context of psoriasis pathogenesis.

## 5. Conclusions

Overall, our study identifies DEGs in mildly and severely inflamed keratinocytes, forming the basis for developing biomarker panels in diagnosing and predicting the course of psoriasis in individual patients, facilitating prognosis and treatment decisions. The research also suggests that epigenetic modifiers play a crucial role in responding to psoriasis-like inflammation, potentially restoring normal gene regulation in keratinocytes. This suggests a potential avenue for future therapies where epigenetic modifications could be targeted, especially for psoriasis patients unresponsive to conventional treatments. In conclusion, further investigations can delve deeper into the roles of specific genes, epigenetic modifiers, and signaling pathways identified in this research, potentially unveiling innovative therapeutic targets and strategies for psoriasis.

## Figures and Tables

**Figure 1 cells-12-02825-f001:**
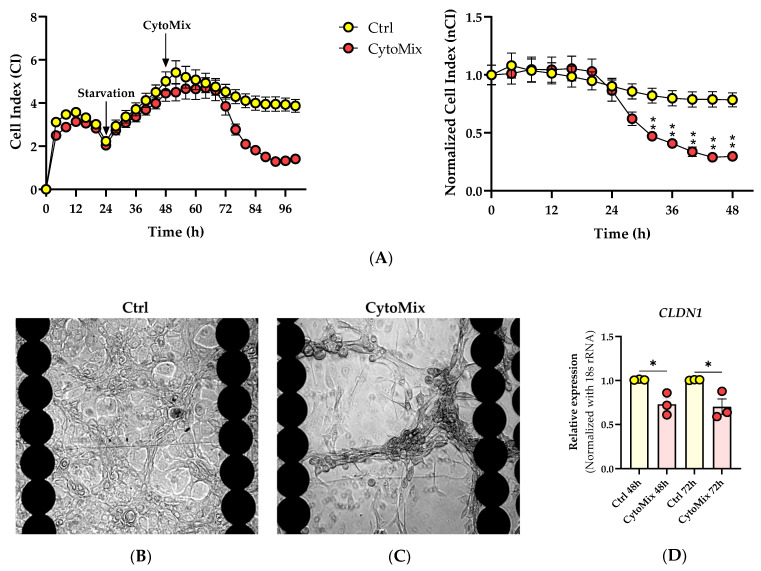
Effects of cytokine mixture on keratinocyte behavior, morphology, and cell–cell junction. (**A**) Analysis of the cellular fate of keratinocytes in response to cytokine mixture treatment using RTCA. Representative CI and nCI graphs are shown (*n* = 3). Each data point represents the mean of three technical replicates ± the standard error of the mean (SEM). (**B**,**C**) Transmitted light microscopy images of CytoMix-treated and control KCs at the end of RTCA experiment. Representative transmitted light microscopy images of the control and CytoMix-treated cultures of KCs in E-plates 96 well are shown (*n* = 3). Scale bars are 500 µm. (**D**) *CLDN-1* gene expression analysis using Real-time RT-PCR. The mRNA expression levels of *CLDN1* gene were normalized using the *18S* rRNA housekeeping gene. The results were quoted as the fold-increase or decrease in expression relative to the time-matched unstimulated controls. Each dot in bar charts represents a biological replicate ± the standard error of the mean (SEM) (*n* = 3). A Student’s unpaired *t*-test was performed for statistical evaluation, and *p*-values were as follows: * *p* < 0.05 and ** *p* < 0.01.

**Figure 2 cells-12-02825-f002:**
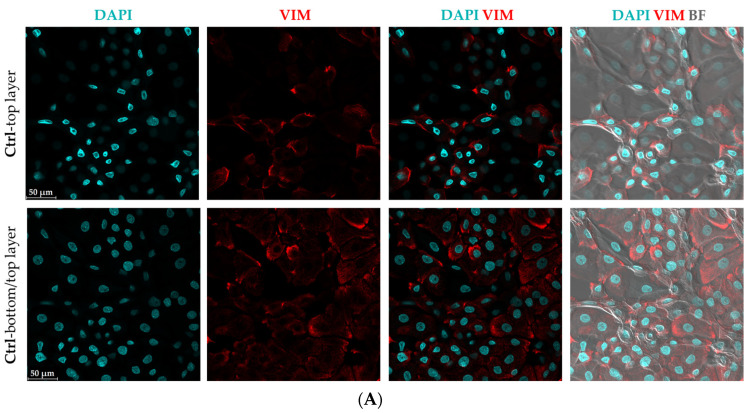
Evaluation of epithelial–mesenchymal transition (EMT) characteristics in cytokine mixture-treated keratinocytes. The representative confocal microscopy images of keratinocytes immunostained for Vim (red) and DAPI (blue) were captured from (**A**) untreated control cells and (**B**) cells treated with CytoMix at 48 h post-treatment (*n* = 3). (**C**) The polygonal KC in control group and spindle-shaped KCs with long-armed intercellular connections in CytoMix-treated group are marked with arrows. Scale bars are 50 μm (in panels (**A**–**C**)). (**D**) Real-time RT-PCR analysis for *VIM* gene expression level was carried out on control and CytoMix-treated KCs at 48 and 72 h post-treatment. The mRNA expression levels of *VIM* were normalized using the *18S* rRNA housekeeping gene. The results were quoted as the fold-increase or decrease in expression relative to the time-matched unstimulated controls. Each dot in bar charts represents a biological replicate ± the standard error of the mean (SEM) (*n* = 3). A Student’s unpaired *t*-test was performed for statistical evaluation, and *p*-values were as follows: *** *p* < 0.001.

**Figure 3 cells-12-02825-f003:**
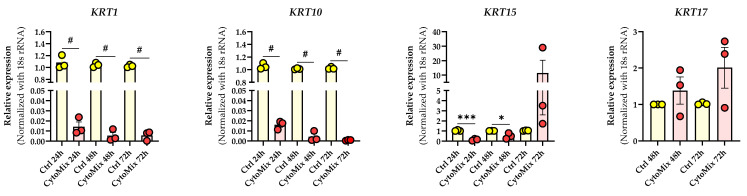
Cytokine mixture treatment effects on psoriasis-associated keratin genes expression. The KCs were treated with or without CytoMix for up to 72 h. Real-time RT-PCR analysis was carried out, and mRNA expression levels for *KRT1*, *KRT10*, *KRT15*, and *KRT17* were normalized using the *18S* rRNA housekeeping gene. *KRT17* mRNA expression was analyzed at 48 and 72 h. The results were quoted as the fold-increase or decrease in expression relative to the time-matched unstimulated Controls. Each dot in bar charts represents a biological replicate ± the standard error of the mean (SEM) (*n* = 3). A Student’s unpaired *t*-test was performed for statistical evaluation, and *p*-values were as follows: * *p* < 0.05, *** *p* < 0.001, # *p* < 0.0001.

**Figure 4 cells-12-02825-f004:**
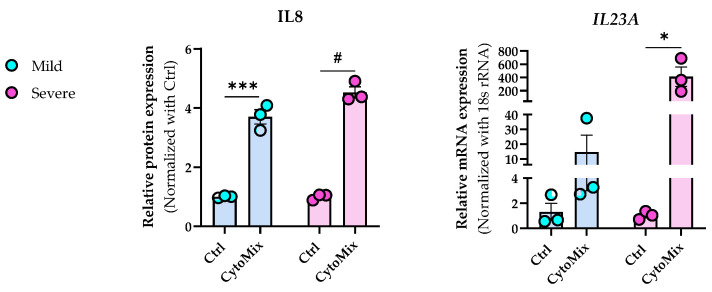
The impact of increased cytokine mixture concentrations (10× CytoMix) on IL-8 protein and IL-23A mRNA expression in KCs. The KCs were treated with or without CytoMix at both basic and 10× higher concentrations of each cytokine/growth factor for 48 h to induce mild and severe inflammation, respectively. IL-8 release in cell culture media was quantified by the ELISA method. The ELISA results were normalized against the control conditions and are presented as fold changes compared to the controls. Real-time RT-PCR analysis was conducted to measure the *IL-23A* mRNA expression levels. The mRNA expression levels for *IL-23A* were normalized using the *18S* rRNA housekeeping gene. The results are expressed as the fold change in expression compared to controls. Each dot in bar charts represents a biological replicate ± the standard error of the mean (SEM) (*n* = 3). A Student’s unpaired *t*-test was performed to evaluate ELISA and real-time RT-PCR experiments for statistical analysis. The *p*-values were as follows: * *p* < 0.05, *** *p* < 0.001, # *p* < 0.0001.

**Figure 5 cells-12-02825-f005:**
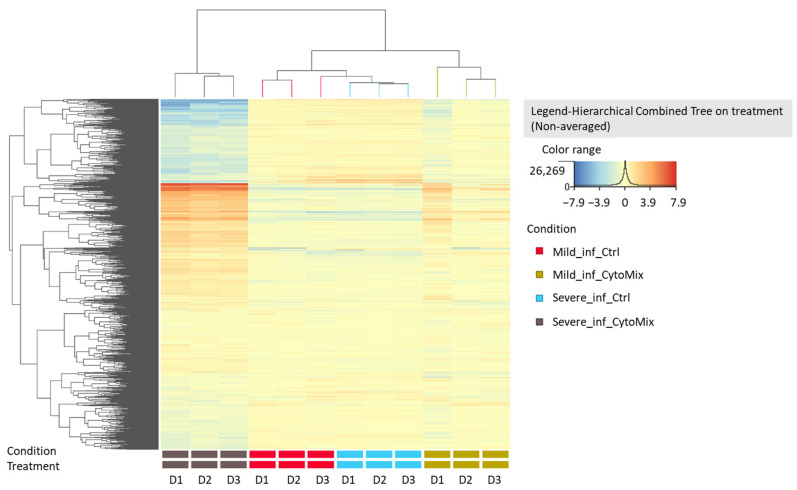
Heatmap analysis of gene expression patterns in mildly and severely inflamed keratinocytes. The heatmap analysis displays the gene expression patterns of mildly and severely inflamed KCs compared to their respective controls at 48 h post inflammation induction (*n* = 3). Red colors represent up-regulated genes, indicating higher expression levels, while blue colors indicate down-regulated genes, reflecting lower expression levels. Abbreviations: Mild_inf_Ctrl, control group of keratinocytes with mild inflammation induction; Mild_inf_CytoMix, keratinocytes treated with mild cytokine mixture; Severe_inf_Ctrl, control group of keratinocytes with severe inflammation induction; Severe_inf_CytoMix, keratinocytes treated with 10× cytokine mixture; D1, biological donor 1.

**Figure 6 cells-12-02825-f006:**
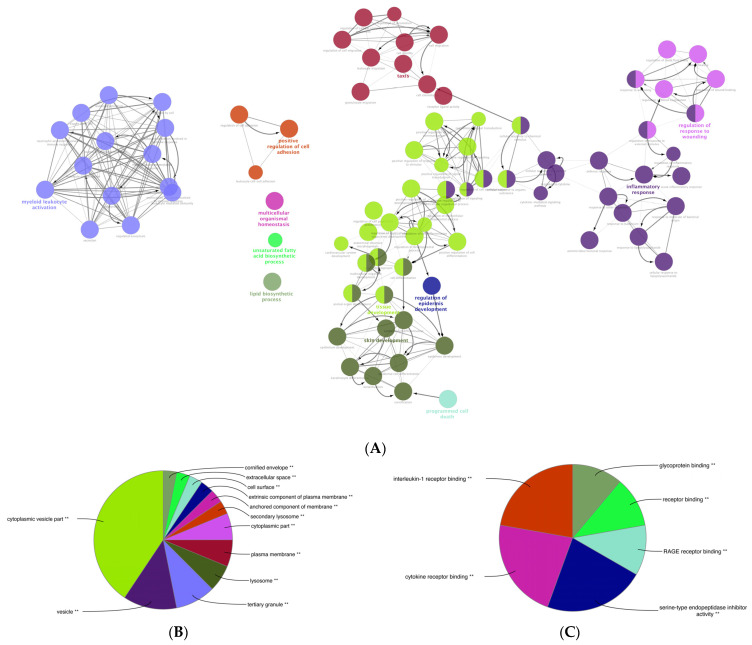
The GO enrichment analysis of the DEGs between mildly (1× CytoMix) inflamed KCs and their controls. Graphs show only significant BPs, CCs, and MFs with *p* < 0.05. In (**A**) BPs graph, the node color corresponds to the specific functional class of each biological process. Bold fonts highlight the major BPs, which serve as defining names for each group. The pie charts display the results of GO analysis for (**B**) CC and (**C**) MF categories. The size of the slices in pie charts correlates with the number of terms grouped together. The ** sign denotes a significant overrepresentation of the presented GO terms on the pie chart, with a corrected *p*-value of < 0.001.

**Figure 7 cells-12-02825-f007:**
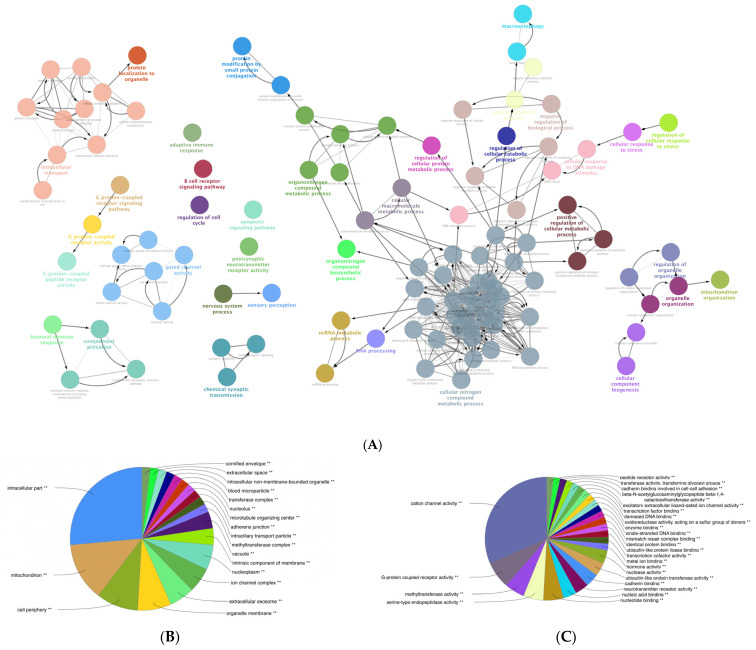
GO enrichment analysis of the DEGs between severely (10× CytoMix) inflamed KCs and their controls. Graphs show only significant BPs, CCs, and MFs with *p* < 0.05. In (**A**) BPs graph, the node color corresponds to the specific functional class of each biological process. Bold fonts highlight the major BPs, which serve as defining names for each group. The pie charts display the results of GO analysis for (**B**) CC and (**C**) MF categories. The size of the slices in pie charts correlates with the number of terms grouped together. The ** sign denotes a significant overrepresentation of the presented GO terms on the pie chart, with a corrected *p*-value of < 0.001.

**Figure 8 cells-12-02825-f008:**
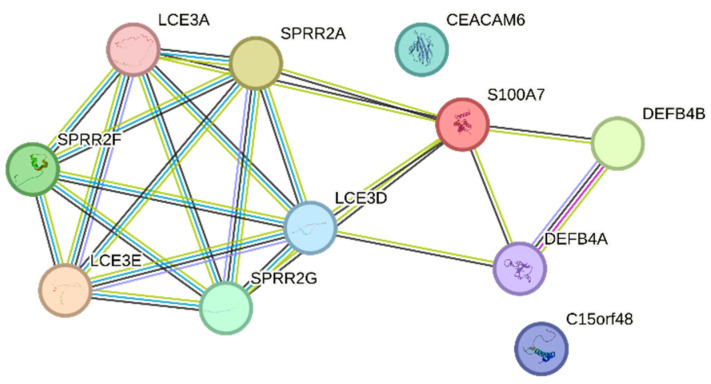
PPI networks of the 12 up-regulated shared genes between mildly and severely inflamed KCs were generated using the STRING database.

**Figure 9 cells-12-02825-f009:**
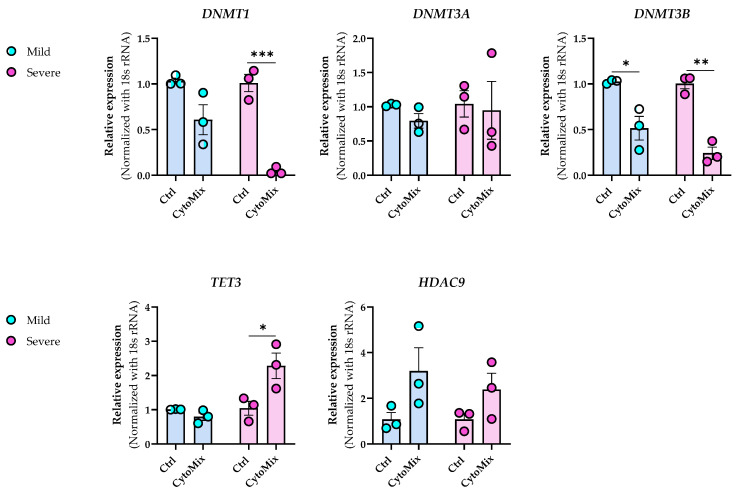
Dynamic epigenetic alterations in HDAC9, DNMTs, and TET3 expression during mild and severe psoriasis-like inflammation in keratinocytes. The mRNA expression levels of *DNMT1*, *DNMT3A*, *DNMT3B*, *HDAC9*, and *TET3* were assessed in the same samples that were used for RNA-sequencing analysis. The expression of these genes was normalized using the *18S* rRNA housekeeping gene. The reported results represent the fold change in expression compared to the corresponding unstimulated controls. Each dot in bar charts represents a biological replicate ± the standard error of the mean (SEM) (*n* = 3). To evaluate statistical significance, a Student’s unpaired *t*-test was conducted, and the resulting *p*-values were indicated as follows: * *p* < 0.05, ** *p* < 0.01, *** *p* < 0.001.

**Figure 10 cells-12-02825-f010:**
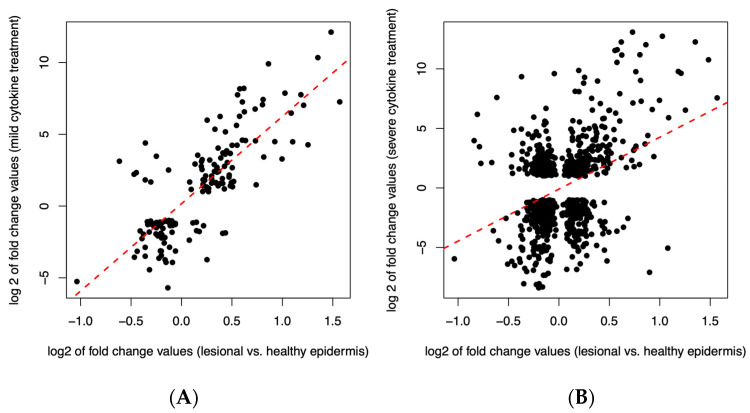
The log2-transformed fold change values of overlapping DEGs are shown for mild cytokine treatment and lesional vs. healthy epidermis ((**A**), *n* = 157) and for severe cytokine treatment and lesional vs. healthy epidermis ((**B**), *n* = 851). Red dashed linear regression lines are also indicated.

**Figure 11 cells-12-02825-f011:**
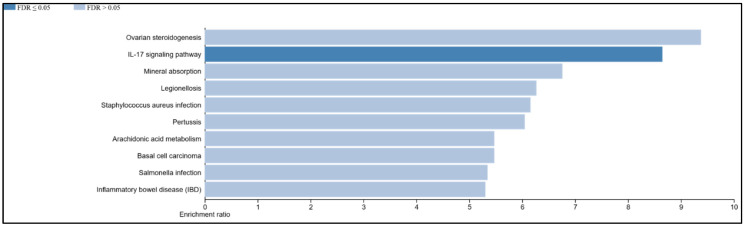
KEGG pathway analysis of overlapping DEGs in mildly inflamed KCs and psoriatic lesional epidermis. In this analysis, 157 genes shared between the DEGs in mildly inflamed KCs and the DEGs in psoriatic lesional epidermis were subjected to KEGG pathway analysis.

**Figure 12 cells-12-02825-f012:**
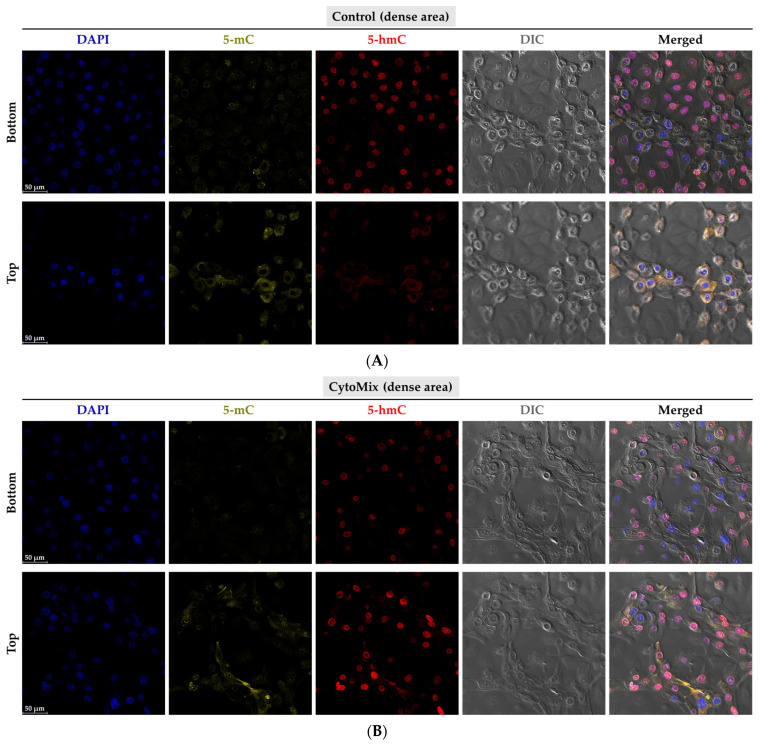
Investigation of 5-mC and 5-hmC patterns, TET gene expression, and 5-mC levels in psoriasis-like inflamed KCs. The KCs were stained 48 h post mild inflammation induction for 5-mC (in yellow), 5-hmC (in red), and DAPI (in blue). The confocal microscopy representative images were taken from the bottom and top layers of (**A**) control and (**B**) CytoMix-treated KCs (*n* = 3). Scale bars are 50 μm (in panels (**A**,**B**)). (**C**) The mRNA expression levels of *TET1*, *TET2*, and *TET3* were normalized using the *18S* rRNA housekeeping gene. The results were quoted as the fold-increase or decrease in expression relative to the time-matched unstimulated controls. (**D**) The 5-mC% of the total DNA was assessed 48 h post mild and severe inflammation induction in KCs using a Global DNA methylation assay kit. Each dot in bar charts and dot graphs represents a biological replicate ± the standard error of the mean (SEM) (*n* = 3). A Student unpaired *t*-test was performed for statistical evaluation, and *p*-values were as follows: * *p* < 0.05, ** *p* < 0.01, *** *p* < 0.001, # *p* < 0.0001.

**Table 1 cells-12-02825-t001:** The DEGs of mildly (1× CytoMix) and severely (10× CytoMix) inflamed KCs vs. their controls.

Samples	DEGs	Down-Regulated	Up-Regulated
Mildly inflamed KCs	858	285 (33.21%)	573 (66.79%)
Severely inflamed KCs (10× CytoMix)	6987	3655 (52.31%)	3332 (47.69%)

**Table 2 cells-12-02825-t002:** The 20 genes displaying the greatest expression changes among the DEGs found in mildly and severely inflamed KCs were sorted according to their fold of change (FC).

**Mildly-Inflamed KCs DEGs (CytoMix-Treated vs. Control)**									
**Down-regulated DEGs**					**Up-regulated DEGs**				
**Number**	**Gene Symbol**	**Description**	**FC**	**Log FC**	**Number**	**Gene Symbol**	**Description**	**FC**	**Log FC**
1	KRT1	keratin 1	−151.88	−7.25	1	S100A7A	S100 calcium binding protein A7A	4453.61	12.12
2	KRT125P	keratin 125 pseudogene	−82.84	−6.37	2	S100A7	S100 calcium binding protein A7	1299.46	10.34
3	KRT10	keratin 10	−62.23	−5.96	3	DEFB4A	defensin beta 4A	963.52	9.91
4	CXCL14	C-X-C motif chemokine ligand 14	−52.00	−5.70	4	DEFB4B	defensin beta 4B	782.22	9.61
5	KRT77	keratin 77	−38.38	−5.26	5	CEACAM6	carcinoembryonic antigen related cell adhesion molecule 6	337.34	8.40
6	DSC1	desmocollin 1	−26.53	−4.73	6	SPRR2C	small proline rich protein 2C (pseudogene)	294.24	8.20
7	CHP2	calcineurin like EF-hand protein 2	−21.70	−4.44	7	C15orf48	chromosome 15 open reading frame 48	293.36	8.20
8	CALML3	calmodulin like 3	−21.13	−4.40	8	SPRR2A	small proline rich protein 2A	287.57	8.17
9	ZBTB16	zinc finger and BTB domain containing 16	−19.90	−4.31	9	SPRR2F	small proline rich protein 2F	287.26	8.17
10	BBOX1	gamma-butyrobetaine hydroxylase 1	−16.34	−4.03	10	CLDN17	claudin 17	246.05	7.94
11	AADACL2	arylacetamide deacetylase like 2	−15.30	−3.94	11	LCE3A	late cornified envelope 3A	233.99	7.87
12	DSG1	desmoglein 1	−15.22	−3.93	12	SPRR2G	small proline rich protein 2G	231.50	7.85
13	POU3F1	POU class 3 homeobox 1	−15.16	−3.92	13	S100A12	S100 calcium binding protein A12	216.71	7.76
14	MYO3B	myosin IIIB	−14.94	−3.90	14	LCE3D	late cornified envelope 3D	215.56	7.75
15	THEM5	thioesterase superfamily member 5	−14.30	−3.84	15	SPRR2B	small proline rich protein 2B	206.64	7.69
16	EPHB6	EPH receptor B6	−13.49	−3.75	16	SLC6A14	solute carrier family 6 member 14	177.68	7.47
17	CCDC3	coiled-coil domain containing 3	−13.43	−3.75	17	IL36G	interleukin 36, gamma	171.82	7.42
18	DNASE1L3	deoxyribonuclease 1 like 3	−13.34	−3.74	18	S100A9	S100 calcium binding protein A9	153.24	7.26
19	FRG2HP	FSHD region gene 2 family member H, pseudogene	−13.21	−3.72	19	RHCG	Rh family C glycoprotein	152.64	7.25
20	PALMD	palmdelphin	−12.44	−3.64	20	LCE3E	late cornified envelope 3E	133.70	7.06
**Severely-Inflamed KCs DEGs (CytoMix-Treated vs. Control)**									
**Down-regulated DEGs**					**Up-regulated DEGs**				
**Number**	**Gene Symbol**	**Description**	**FC**	**Log FC**	**Number**	**Gene Symbol**	**Description**	**FC**	**Log FC**
1	ID3	inhibitor of DNA binding 3, HLH protein	−571.72	−9.16	1	DEFB103B	defensin beta 103B	8680.30	13.08
2	DSC1	desmocollin 1	−487.55	−8.93	2	LCE3A	late cornified envelope 3A	6831.53	12.74
3	EPHB6	EPH receptor B6	−429.75	−8.75	3	DEFB103A	defensin beta 103A	6802.54	12.73
4	APCDD1	APC down-regulated 1	−335.49	−8.39	4	S100A7	S100 calcium binding protein A7	4901.32	12.26
5	ECM2	extracellular matrix protein 2	−333.54	−8.38	5	SPRR2C	small proline rich protein 2C (pseudogene)	4883.04	12.25
6	CCDC3	coiled-coil domain containing 3	−318.71	−8.32	6	DEFB4A	defensin beta 4A	4175.80	12.03
7	PALMD	palmdelphin	−277.46	−8.12	7	CXCL9	C-X-C motif chemokine ligand 9	3962.80	11.95
8	KRT1	keratin 1	−277.14	−8.11	8	DEFB4B	defensin beta 4B	3775.58	11.88
9	TXNIP	thioredoxin interacting protein	−264.49	−8.05	9	CEACAM6	carcinoembryonic antigen related cell adhesion molecule 6	3679.34	11.85
10	EPHX2	epoxide hydrolase 2	−264.19	−8.05	10	SPRR2A	small proline rich protein 2A	3117.66	11.61
11	KLRG2	killer cell lectin like receptor G2	−264.06	−8.04	11	ACP7	acid phosphatase 7, tartrate resistant (putative)	2982.39	11.54
12	LYRM7	LYR motif containing 7	−262.17	−8.03	12	LCE3D	late cornified envelope 3D	2980.27	11.54
13	IGFL2	IGF like family member 2	−250.56	−7.97	13	SPRR2G	small proline rich protein 2G	2869.05	11.49
14	PIR	pirin	−240.98	−7.91	14	CRCT1	cysteine rich C-terminal 1	2807.03	11.45
15	RAB7B	RAB7B, member RAS oncogene family	−240.34	−7.91	15	C15orf48	chromosome 15 open reading frame 48	2718.28	11.41
16	IL20RA	interleukin 20 receptor subunit alpha	−212.41	−7.73	16	SPRR2F	small proline rich protein 2F	2608.19	11.35
17	CYP39A1	cytochrome P450 family 39 subfamily A member 1	−198.56	−7.63	17	IGFN1	immunoglobulin-like and fibronectin type III domain containing 1	2348.78	11.20
18	CCDC152	coiled-coil domain containing 152	−181.65	−7.50	18	LCE3E	late cornified envelope 3E	2325.20	11.18
19	FLG	filaggrin	−181.30	−7.50	19	PRSS22	protease, serine 22	2291.30	11.16
20	S1PR5	sphingosine-1-phosphate receptor 5	−180.17	−7.49	20	MMP10	matrix metallopeptidase 10	2194.50	11.10

**Table 3 cells-12-02825-t003:** Alterations in epigenetic modifier mRNA expression in mildly and severely inflamed keratinocytes based on our RNA-sequencing data.

Gene Symbol	Mildly Inflamed KCs vs. Controls		Severely Inflamed KCs vs. Controls	
	Fold of Change (FC)	*p*-Value	Fold of Change (FC)	*p*-Value
** *HAT1* **	−1.10	0.6981141	−2.45	2.708402 × 10^−4^
** *HDAC9* **	+4.60	0.035447594	+7.40	1.0198189 × 10^−4^
** *DNMT1* **	−1.31	0.017961252	−7.30	0.0012681335
** *DNMT3A* **	−1.06	0.52116096	+2.24	3.4366465 × 10^−5^
** *DNMT3B* **	−2.20	0.22995375	−1.38	0.04189076
** *TET3* **	−1.25	0.5099057	+3.97	3.0937628 × 10^−4^

## Data Availability

Data are presented in the manuscript and are available upon request from the corresponding author.

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
