# Peer review of "Unraveling Transcriptome Profile, Epigenetic Dynamics, and Morphological Changes in Psoriasis-like Keratinocytes: “Insights into Similarity with Psoriatic Lesional Epidermis”"

_cells, 2023, doi:10.3390/cells12242825_

Round 1

Reviewer 1 Report

Comments and Suggestions for Authors

Please find my comments below.

Line 28. “We created a novel cytokine mixture to mimic mild and severe psoriasis-like inflammatory conditions in cultured keratinocytes”. – please provide the detailed description of the mixture composition

Line 131. “n.2.3. xCELLigence real-time cell analysis (RTCA)” please fix the formatting issue

Line 496. Please fix the formatting issue (number 496 overlaps with the text in the table)

Can the authors elaborate more on discussing the direct clinical/preclinical application of their work?

What about the racial (and perhaps ethnic-specific) differences in the response of skin to pro-inflammatory cytokines (for example, see Budunova I et al, Differential response of 3D African American and White Non-Hispanic skin organoids to major pro-inflammatory cytokines. Journal of Investigative Dermatology. 2023 May 1;143(5):S225)?

The article is a high quality manuscript and in my opinion it should be published in its current state.

Could you please provide more info about racial/ethnic composition of the donor’s cohort? The study was performed in Hungary, but donor’s cohort may be diverse in terms of racial/ethnic composition. How many samples the donor’s cohort included? Were there any significant differences / donor-dependent variations in terms of cells response to the cytokines?

Author Response

Dear Reviewer,

Thank you for your valuable feedback on our manuscript. We appreciate the time and effort you have dedicated to the review process. In this response letter, we have addressed your comments and suggestions, and we hope that our revisions have successfully addressed your concerns. Please find our point-by-point responses in the attached PDF file.

Best Regards, 

Lajos Kemény

Reviewer 2 Report

Comments and Suggestions for Authors

The paper by Ghaffarinia and Coll deals with the interesting issue of the impact of proinflammatory psoriatic cytokines on the target cytotype of this disease, i.e. the keratinocytes. In particular, I appreciate the addition of KGF to the inflammatory cocktail, because I think that its contribution can be relevant. The Results herein presented are potentially interesting, but their presentation must be improved as many infos are dispersed throughout the manuscript. First of all, the Figure Legends should refer only to the description of the results and any further comment must be included in the Result Section.

In the Mat/Met Section

- In the paragraph 2.2 the definition of CI and nCI must be better explained

- lines 137- 141 are repeated in the Results (lines 270-274): please correct and, consequently, move Figure 1A in Mat/Met Section

In the Result Section

- Please rename the panels in Figure 1

- Figure 1c is not clear. Please provide better photomicrographs

- Lines 324-327, 430-434, 440-442, 444-446, 459-462, 573-576, 663-668, 675-677, 680-682 do not refer to Results, but must be inserted either in the Intro or in the Discussion Section

- The difference among Figure 2A, 2B, and 2C is not clear.

- Table 3 is not mentioned in the text

- Lines 620-625 but must be inserted in the Mat/Met Section

In the Discussion Section

I don't agree with the statement at line 716-717 "simulating psoriasis pathogenesis", because it's underestimating the complexity of this issue. I strongly encourage the Authors rephrasing this sentence considering the huge amount of recent literature concerning the pathogenesis of psoriasis.

Lines 734-737: Authours should extrapolate this consideration to clinics. Which is the relevance for of EMT in clinical practice?

Line 746: I strongly suggest a proliferation assay for a correlation with K15 expression

Lines 765-767: Authors should discuss the relevance of Th17 pathway in psoriasis in more detail, with a particular focus to K17 expression

Author Response

(The authors gave the same response as above.)

Reviewer 3 Report

Comments and Suggestions for Authors

Author Response

(The authors gave the same response as above.)

Round 2

Reviewer 2 Report

Comments and Suggestions for Authors

I appreciate the graphical abstract provided by the Authors that summarizes their experimental plan. Conversely, I'm not convinced of the quality of the presentation of the results as they're confusing, especially in the first part, and I still have the following major concerns: 

- The explanation for CI and nCI is now clear, but it's not clear which is its relevance, i.e. what do Authors measure by this parameter? What do they mean by "cellular behavior"? It's quite an innovative technique and the reader should be better informed.

- Overall, I'm still convinced that Figure 1A is not pertinent to the Result Section, but should either be moved in the Mat/Met Section or deleted.

- In Paragraph 3.1, Figures should be mentioned/described in order of appearance, and lines 275-276 are inappropriate in this section, but should belong to Introduction

- The legend of Figure 1 is not homogeneous if compared with all other legends as it describes and comments on the results.

- In Paragraph 3.2 lines 319-323 are inappropriate as they refer to the background, so they must be placed in the Introduction.

- I don't find, in the Discussion Section, the appropriate consideration regarding Figures 2A, 2B, and 2C

- In the legend of Figure 2, the time point considered is lacking

- In Figure 4, the explanation of mild/severe should be added in the legend as each Figure should stand alone and the reader should be able to comprehend without consulting the main text

The sentence in lines 731-733 is too general for the requested extrapolation to the clinics.

Author Response

Dear Reviewer,

We sincerely appreciate your thoughtful comments and helpful feedback on our manuscript. Your input during the first and second reviews has been invaluable in enhancing the quality and clarity of our work. Please find a detailed, point-by-point response to your comments in the enclosed PDF file. We hope that the updated version now addresses the concerns efficiently.

Sincerely yours,

Lajos Kemény
